# Fluoride Exposure from Soybean Beverage Consumption: A Toxic Risk Assessment

**DOI:** 10.3390/foods11142121

**Published:** 2022-07-17

**Authors:** Virginia Mesa-Infante, Daniel Niebla-Canelo, Samuel Alejandro-Vega, Ángel J. Gutiérrez, Carmen Rubio-Armendáriz, Arturo Hardisson, Soraya Paz

**Affiliations:** 1Área de Toxicología, Universidad de La Laguna, 38071 La Laguna, Tenerife, Spain; alu0101004389@ull.edu.es (V.M.-I.); alu0100798203@ull.edu.es (D.N.-C.); alu0100992397@ull.edu.es (S.A.-V.); ajguti@ull.edu.es (Á.J.G.); crubio@ull.edu.es (C.R.-A.); atorre@ull.edu.es (A.H.); 2Grupo Interuniversitario de Toxicología Alimentaria y Ambiental, Universidad de La Laguna, 38071 La Laguna, Tenerife, Spain

**Keywords:** fluoride, soybean beverages, risk assessment, human exposure, plant-based alternative

## Abstract

The consumption of vegetable milk as a substitute for cow’s milk has increased in recent years. Of all the vegetable beverages on the market, soy is the most widely consumed. Soy is exposed to contamination by different chemical elements during harvesting. In this study, the concentration of fluoride in soy beverages was analyzed. Fluoride is an element that in high concentrations can be toxic, causing dental and bone fluorosis. The aims of the study were (i) to analyze the fluoride concentration in 30 samples in the most popular brands (A-Brand, B-Brand, C-Brand) of soybean beverages by a fluoride ion selective potentiometer and (ii) to evaluate the toxicological risk derived from its consumption. The fluoride concentrations were 15.5 mg/L (A-Brand), 11.3 mg/L (B-Brand) and 8.5 mg/L (C-Brand). A consumption of 1 to 3 servings/day was established. One serving (200 mL) of soybean beverage offered a contribution percentage over the ADI (acceptable daily intake) for infants and children. Teenagers and adults did not exceed the ADI (10 mg/day). The consumption of soy beverages contributes significantly to the daily intake of fluoride, which could exceed the ADI with a consequent health risk. It is recommended to control the fluoride levels in the raw material and in the final product to assure the safety of these products.

## 1. Introduction

Vegetable beverages have been consumed throughout history as substitutes for cow’s milk. Vegetable beverages are extracts from legumes, seeds, cereals, etc. in water whose appearance resembles cow’s milk. There is a great variety, although soy drinks are the most consumed [1]. Consumption and production of vegetable beverages have been increasing. According to an estimation made at the 38th National Congress of the Spanish Society of Primary Care Physicians (SEMERGEN) in 2016, global consumption of vegetable beverages will increase by 2.2%. However, in countries such as Spain, the increase is expected to be 15.8% [2].

The reasons why the consumption of vegetable beverages such as soybean beverages is increasing is mainly due to a rejection of dairy products due to lactose intolerance, allergy to cow’s milk proteins and/or cholesterol problems [3]. In addition, in recent years, vegetable and plant sources (cereals and legumes) have been accepted as functional foods due to the presence of components that promote health benefits, which is favoring an increase in its consumption [4].

Soybean beverages, compared to other vegetable beverages, are most similar in terms of nutritional value to cow’s milk [5]. However, cow’s milk is 15% and 10% richer in branched chain amino acids (BCAAs) and essential amino acids (EAAs), respectively. Although it is a good source of fiber, minerals such as Fe or Zn and certain vitamins, the bioavailability of calcium is lower, and it is not a good source of vitamin B12, among other vitamins [6,7]. Therefore, milk for infants or children should be supplemented [8].

Soybean is rich in polyunsaturated fatty acids (n-3 and n-6). Studies in experimental animals indicate that diets containing soybean beverages favor a reduction in high-density cholesterol (HDL) [9]. It also contains potentially bioactive compounds such as phytoestrogens, which are associated with reducing the rates of breast, ovarian, prostate, colon cancer, etc. [10,11]. Although there are studies that link a high consumption of soybean with the risk of developing breast cancer in women with menopause [12], recent studies indicate that it reduces the risk and even improves the symptoms of menopause [13,14,15].

The increase in environmental pollution and its impact on crops, especially soybeans, constitutes a risk to consumer health. Soy is capable of absorbing potentially toxic elements [16,17,18]. Among the different chemical elements that can pose a health risk, it is worth highlighting the fluoride ion. Fluorine belongs to the group of halogens and is the most electronegative element in the periodic table, which gives it great chemical reactivity. The fluoride ion is widely distributed in nature. The main component responsible for its contribution to the diet is water or foods reconstituted with water, such as soup, infant formula and vegetable drinks [4]. The concentration of this ion can increase for various reasons such as human activity (pesticides, water fluoridation, etc.) or natural phenomena (volcanic emissions, soil erosion with high fluoride content, fluoride leaching into water) [18]. In the case of soybean beverages, the reasons for the presence of fluoride are the fluoride content of the soil, the fluoride concentration of the water used in the soybean harvest and the water used in producing the homogenate.

The European Food Safety Authority (EFSA) indicates that it is a non-essential trace element since it is not essential [19,20]. The main route of fluoride absorption is passively through the gastrointestinal tract. The fluoride distribution through the blood facilitates its deposit mainly in the teeth, favoring the formation of fluorapatite and giving greater strength to dental enamel and resistance to acids and inhibiting the metabolism of cariogenic bacteria. Likewise, it increases its hardness in the bones and prevents the appearance of osteoporosis.

However, if fluoride intake is excessive (>10 mg/L day), it is related to dental fluorosis characterized by the appearance of opacity, discoloration of the dental enamel and disfigurement of the teeth. Furthermore, bone fluorosis can occur as a consequence of an increase in bone mass and ossification of cartilage due to alteration of calcium metabolism. In addition, excess is related to metabolic, brain, fertility problems, etc. [21,22,23,24,25]. For this reason, the Institute of Medicine’s Food and Nutrition Board [26] established acceptable daily intake (ADI) levels based on age and physiological state (Table 1).

Soybean beverages do not have maximum fluoride levels in European or North American legislation; therefore, due to the lack of regulation of the content of this ion, the value established for water for human consumption will be taken as an example, which has a parametric level of 1.5 mg/L of fluoride [27,28]. Likewise, it should be noted that in bottled water, it is mandatory to indicate the level of fluoride [29]. 

These data and studies previously published by various authors [30,31,32] indicate that soybean beverages may have a high content of fluoride, favoring dental fluorosis as a consequence of use of water for its manufacture. Thus, the need to determine the fluoride content in marketed soy beverages is established.

The objectives established are (i) to determine the levels of fluoride in soybean beverages of the most popular commercial brands in Spain, (ii) to study the possible statistical differences between the soybean beverages of the different brands analyzed and (iii) to evaluate the toxic risk from fluoride intake as a consequence of the consumption of soybean beverages, considering the values of admissible daily intake (ADI).

## 2. Materials and Methods

### 2.1. Materials and Solutions

Milli-Q quality ultrapure distilled water, obtained from an ultrafiltration system (Milli-Q^®^ Direct, Merck KGaA, Darmstadt, Germany), was used in all cases. Since fluoride interacts with borosilicate in glass, the use of glass material was avoided at all times, using sterile plastic containers, plastic spatulas and plastic volumetric flasks with a nominal capacity of 25 mL.

A stock solution of fluoride with a concentration of 10^−1^ M (mol/L) was prepared, dissolving 0.42 g of sodium fluoride (NaF) (Sigma Aldrich, Schnelldorf, Germany), previously dried in an oven (Nabertherm, Lilienthal, Germany) for 2 h at 120 °C.

The buffer solution used was a 0.75 M orthophosphoric acid (H_3_PO_4_) buffer (Sigma Aldrich, Germany), which was prepared by dissolving 51 mL of 85% orthophosphoric acid in 1 L of distilled water. The solution was labeled and stored in a dark-colored container.

### 2.2. Samples

The three most popular commercial brands of soybean beverage that could be purchased from the most common large supermarkets were chosen. Specifically, a total of 30 samples of A-Brand, B-Brand and C-Brand, from different batches, were taken (Table 2).

### 2.3. Fluoride Determination

The method used is based on determining the fluoride concentration using a fluoride selective electrode. The electrode consists of a membrane of lanthanum fluoride (LaF_3_) treated with europium fluoride (EuF_2_) to create empty networks, which cause a potential difference depending on the concentration of fluoride in the solution [33]. Because the selective electrode measures the potential difference, the pH and ionic strength of the sample must be adjusted, as the electrode working conditions require a pH from 4 to 8. For the latter, 0.75 M orthophosphoric acid was used as a buffer solution, which is the buffer solution that provided the best results in previous publications [32,33,34]. A potentiometer (HACH, LZ55C.97.002F, Düsseldorf, Germany) was used with pH electrodes (CRISON 5204, Alella, Spain) and a fluoride selective ion electrode (HACH, Düsseldorf, Germany).

First, the equipment was calibrated with standard solutions (HACH, Düsseldorf, Germany). Subsequently, serial dilutions (10^−2^ to 10^−5^ M concentrations) were prepared from the 10^−1^ M fluoride stock solution and the potential of each was measured in triplicate. With the measurements of the potentials, a standard curve of semi-logarithmic nature was constructed (Figure 1), extracting the equation of the line from which the fluoride content in the samples to be analyzed would be extrapolated. In all cases, correlation coefficients of greater than 0.989 were obtained.

### 2.4. Method Quality Control and Validation

Method precision was assessed under repeatability and reproducibility conditions. The test was conducted over 3 alternative days, 15 times on each day. The results from the statistical analysis showed no significant differences (*p* < 0.05). A recovery rate study was conducted to assess the method accuracy, which consisted of the addition of a known amount of fluoride to the samples once their concentration was determined.

The mean recovery rate was more than 99% with a repeatability RSD (relative standard deviation) value of 0.60% and reproducibility of 2.50%. Therefore, the precision and accuracy values were satisfactory as the RSD values were lower than 10% and the recovery rates were high.

### 2.5. Statistical Analysis

Statistical analysis was performed using GraphPad Prism 9.0.2 software (GraphPad Prism, San Diego, CA, USA). Statistical analysis was carried out to study possible significant differences between the three brands.

First, we verified whether the values of the analyzed samples followed a normal distribution by applying the Shapiro–Wilk test [35]. However, since the data did not follow a normal distribution, the non-parametric statistical Kruskal–Wallis test was applied [36]. Values of *p* < 0.05 are considered significant differences.

### 2.6. Dietary Intake Calculations

The risk assessment for fluoride intake is based on calculating the estimated amount of fluoride that is ingested with a serving of soybean beverage. A glass of soybean beverage (200 mL) was considered to be a serving. The estimated daily intake (EDI) is obtained as shown in Equation (1). As for the comparison with the ADI, it is carried out as shown in Equation (2).
EDI (mg/day) = Mean consumption (L/day) × Fluoride concentration (mg/L)(1)
ADI (%) = [EDI/(ADI value)] × 100(2)

When the contribution percentages are greater than 10%, it is considered a significant contribution to the diet, since it is the contribution of a single food. Those percentages greater than 50% may already represent a significant risk. However, when the percentage of contribution to the ADI is close to or greater than 100%, the risk is extreme as long as the consumption is prolonged in time.

## 3. Results and Discussion

### 3.1. Fluoride Concentration in Soybean Beverages

Table 3 shows the mean concentrations (mg/L), the standard deviations (SD) and the minimum and maximum values recorded for the analyzed samples.

As shown in Table 3, C-Brand has the lowest average concentration of fluoride (8.5 mg/L) compared to B-Brand (11.3 mg/L) and A-Brand (15.5 mg/L). Although there are multiple factors that can influence the fluoride content, it should be noted that C-Brand is the only one of the brands that does not include sea salt in its list of ingredients. It is known that sea salt can contain fluoride impurities associated with sodium chloride, which is why this can increase the fluoride content in B-Brand and A-Brand. A study carried out by Wang et al. [37] on salt from China showed that sea salt (0.130 mg/kg) registered a higher concentration of fluorides compared to salt of non-marine origin (0.09 mg/kg). However, the statistical analysis carried out has confirmed that there are no significant differences (*p* > 0.05 in all cases) in the fluoride content between the different brands analyzed.

Based on the results obtained, the fluoride levels in the brands analyzed are higher than the parametric values allowed in RD 140/2003 [28] for water for human consumption and the value recommended by [27] in water (1.5 mg/L). Another noteworthy fact is the high deviations registered between samples of the same brand. This highlights the lack of regulation in the fluoride content of both ingredients and raw materials, as well as in the final product. Considering the toxicological importance of fluoride, it is necessary that the competent authorities consider the establishment of legal limits of fluoride in this type of product.

### 3.2. Comparison with Other Authors

Table 4 collects data published by other authors who have analyzed the fluoride content in soybean beverages.

A study carried out in Thailand by Rirattanapong et al. [32] recorded fluoride content of 0.01–3.49 mg/L in 25 brands analyzed. In another study carried out in England, fluoride concentrations were even lower (0.015–0.964 mg/L) [38]. On the other hand, in a study where the fluoride concentration in powdered soybean beverages for infants was analyzed, it showed lower values. The samples of powdered soybean beverages for infants analyzed by de Carvalho et al. [30] contained a fluoride concentration between 0.03 and 0.50 mg/L. Studies carried out by Nagata et al. [31] registered higher fluoride concentrations in soybean beverage lyophilizates compared to the rest of the authors, as is the case of the SoyMilke brand (7.18 mg/L). In general, the concentrations recorded in the present study (2022) are higher than previously published data. 

These concentration differences may be due to multiple factors such as the soybeans used, the water used to make the soybean beverages or the presence of other ingredients. However, it highlights the need to establish raw material traceability programs and fluoride content controls.

### 3.3. Dietary Intake Assessment

At present, the average consumption of soybean beverages is unknown. Although the recommended consumption of dairy products is 2–3 servings a day with portions of 200–250 mL [39,40], the intake of fluorides from the consumption of soybean beverages was estimated. Table 5 shows the estimated daily intake (mg/day) of fluoride and the percentage of contribution to the ADI for each age group, considering 1–3 servings (1 serving = 200 mL of soybean beverage).

In the case of the Acceptable Daily Intake (ADI), infants and children (0–8 years) exceed the ADI value (0.7 to 1.3 mg/day) after consuming just one serving of A-Brand, B-Brand and C-Brand. In the case of children from 4 to 8 years old only, the ADI is not exceeded when 200 mL of C-Brand is consumed, but the percentage of contribution is very high (77.6%), which represents a risk if other dietary fluoride contributions are considered, such as water, tea or coffee. Considering these contribution percentages, it would be advisable to avoid the consumption of soybean beverages in the population group aged 0 to 8 years.

Regarding the consumption of one serving of any of the brands analyzed by children over 9 years of age and the adult population, it does not pose a health risk, but it is necessary to consider that the percentage of contribution is significant (31.0%) and that other dietary sources of fluoride could increase the global intake with a consequent risk to health, leading to the appearance of dental or bone fluorosis, especially in young children, and neurological, heart and fertility problems, among others [24,41,42]. In all cases, moderate consumption should be recommended, since in consumption scenarios of two to three servings, the value of the ADI is already exceeded or close to 100%, depending on the age group.

## 4. Conclusions

The three brands of soybean beverages analyzed have a high concentration of fluoride, especially A-Brand. It should be noted that A-Brand and B-Brand contain sea salt, which can increase the fluoride content in the final product. The consumption of any of these three brands of soybean beverages can pose a health risk in infants and children due to the high amount of fluoride. Intake of up to three servings of soybean beverages in >9 years does not pose a health risk, but considering other dietary sources of fluoride, it may exceed the ADI with consequent health risk. The consumption of one serving is recommended for people over 9 years old. In the case of higher intake, it is advisable to consume the brand with the lowest concentration of fluoride, avoiding consuming more than three servings so as not to exceed the ADI. 

The traceability of the raw material and the control of the fluoride content of the different ingredients that these beverages contain is considered as something completely necessary by the food industry. Likewise, considering the importance of vegetable beverages, such as soybean, it is necessary to establish controls over the fluoride content in these products to ensure their quality and safety.

## Figures and Tables

**Figure 1 foods-11-02121-f001:**
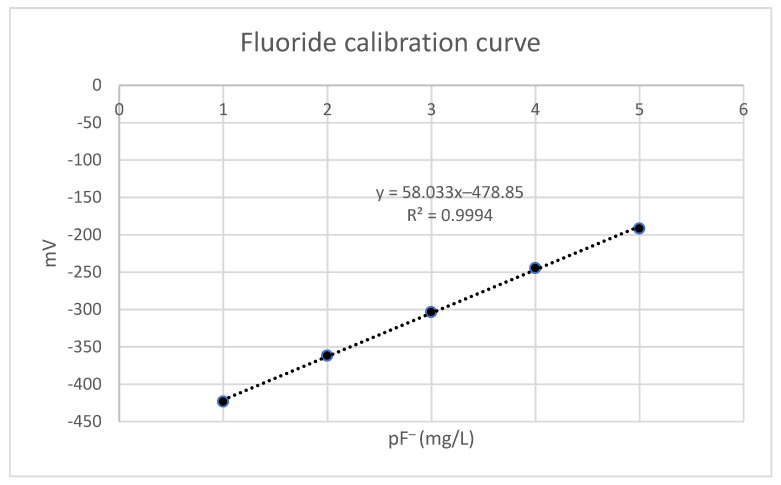
Example of fluoride calibration curve.

**Table 1 foods-11-02121-t001:** Fluoride ADI values (estimated daily intake) according to age and physiological state [26].

Group	Age	ADI (mg/Day)
Infants	0–6 month	0.7
	7–12 month	0.9
Children	1–3 years	1.3
	4–8 years	2.2
	9–13 years	10
Teens	14–18 years	10
Adults	>19–70 years	10
	>70 years	10
Pregnancy	14–50 years	10
Lactation		10

**Table 2 foods-11-02121-t002:** Characteristics of the analyzed samples.

Trademark	No. Samples	Origin	Package Type	Ingredients
A-Brand	10	Spain and other EU countries, Canada	Tetrabrik, 1 L	Soy base (water, shelled soybeans, 8%), sugar, tricalcium phosphate, correct acidity (potassium phosphate), sea salt, flavorings, stabilizer (gellan gum), vitamins (B2, B12, D2)
B-Brand	10	Spain		Natural Park water, 13.5% soy and sea salt
C-Brand	10	Spain and other EU countries, Canada		Water, soybeans (13%), tricalcium phosphate, stabilizers (cellulose and carboxymethylcellulose), flavorings, salt, antioxidants (extract rich in tocopherols, vitamins A and D)

**Table 3 foods-11-02121-t003:** Mean concentrations (mg/L) of fluoride, standard deviations (SD) and minimum and maximum values.

Brand	[F^−^] Mean (mg/L)	SD	Min. Value	Max. Value
C-Brand	8.5	4.8	2.5	20.0
B-Brand	11.3	17.0	2.5	76.2
A-Brand	15.5	19.7	1.0	65.8

**Table 4 foods-11-02121-t004:** Comparison of the fluoride concentrations obtained by other authors.

Sample Type	Brand	Fluoride Concentration (mg/L)	References
Lactose-free soy beverage powder	SupraSoy	0.36	[31]
Unsweetened soy beverage powder	SoyMilke	7.18	
Soy beverage infant formula	Nestlé	0.43	
Soy vegetable beverage	-	0.01–3.49 0.015–0.964	[32,38]
Soya powder beverage, infant formula	-	0.03–0.50	[30]
Ready-to-drink soy beverage	C-Brand	2.5–20	Present study, 2022
	B-Brand	2.5–76.2	
	A-Brand	1.0–65.8	

**Table 5 foods-11-02121-t005:** EDI values (mg/day) and percentage of contribution to the ADI for each age group considering the consumption of 1–3 portions of soybean beverages.

Group	Trademark					
	A-Brand		B-Brand		C-Brand	
	1 Portion (200 mL)					
	EDI (mg/Day)	% ADI	EDI (mg/Day)	% ADI	EDI (mg/Day)	% ADI
0–6 months	3.1	442	2.3	323	1.7	244
7–12 months		344		251		190
1–3 years		238		174		131
4–8 years		141		103		77.6
9–13 years		31.0		22.6		17.1
14–18 years		31.0		22.6		17.1
>19–70 years		31.0		22.6		17.1
**Group**	**2 Portions (400 mL)**					
	**EDI (mg/Day)**	**% ADI**	**EDI (mg/Day)**	**% ADI**	**EDI (mg/Day)**	**% ADI**
0–6 months	6.2	885	4.5	646	3.4	488
7–12 months		688		503		380
1–3 years		476		348		263
4–8 years		282		206		155
9–13 years		61.9		45.2		34.2
14–18 years		61.9		45.2		34.2
>19–70 years		61.9		45.2		34.2
**Group**	**3 Portions (600 mL)**					
	**EDI (mg/Day)**	**% ADI**	**EDI (mg/Day)**	**% ADI**	**EDI (mg/Day)**	**% ADI**
0–6 months	9.3	1327	6.8	969	5.1	732
7–12 months		1032		754		569
1–3 years		715		522		394
4–8 years		422		309		233
9–13 years		92.9		67.9		51.2
14–18 years		92.9		67.9		51.2
>19–70 years		92.9		67.9		51.2

## Data Availability

Data is contained within the article.

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
