# Peer review of "Fluoride Exposure from Soybean Beverage Consumption: A Toxic Risk Assessment"

_foods, 2022, doi:10.3390/foods11142121_

Round 1

Reviewer 1 Report

The manuscript entitled “Fluoride exposure from soybean beverages consumption: A toxic risk assessment “ reveals important scientific results about  fluoride content in soybean beverages. Despite the small number of tested brands, what the authors should take into account in future research, the article is worth reading. The topic is part of the current trends in consuming an increasing amount of cow's milk substitutes, including soy milk. The results of the work are interesting, especially in the context of children, which may affect the attempt at legal regulations in this area.  However, the minor revision should be taken (Line 20 – delete double spaces; Line 108 – should be: 120°; Line 113 – should be 2.2. Samples; Line 123 – delete double spaces; Line 174 - add spaces; Line 184 – should be: Present study, 2022).

Author Response

Comment: The manuscript entitled “Fluoride exposure from soybean beverages consumption: A toxic risk assessment “reveals important scientific results about fluoride content in soybean beverages. Despite the small number of tested brands, what the authors should take into account in future research, the article is worth reading. The topic is part of the current trends in consuming an increasing amount of cow's milk substitutes, including soy milk. The results of the work are interesting, especially in the context of children, which may affect the attempt at legal regulations in this area. 

Response: Authors thank the reviewer for their interest, kind words and suggestions.

Comment: Line 20 – delete double spaces

Response: The typo has been corrected.

Comment: Line 108 – should be: 120°

Response: The change has been done.

Comment: Line 113 – should be 2.2. Samples

Response: The typo has been corrected.

Comment: Line 123 – delete double spaces

Response: The double spaces have been removed.

Comment: Line 174 - add spaces

Response: Space has been added.

Comment: Line 184 – should be: Present study, 2022

Response: The date has been added to all “present study” references.

Reviewer 2 Report

The authors reported here the concentrations of fluoride in soybean beverages belonging to 3 different brands and highlighted the possible fluoride exposure from the consumption of these soybean beverages. After carefully reading it, the reviewer thinks it could be minor revision.

1.     According to table 4, it is obvious that the concentrations of fluoride obtained in this work are higher than those reported previously. Therefore, it would be better to confirmed the results with other techniques.

2.   There are some typos. 

Author Response

Authors thank the reviewer for their suggestions and interest.

Comment: According to table 4, it is obvious that the concentrations of fluoride obtained in this work are higher than those reported previously. Therefore, it would be better to confirmed the results with other techniques.

Response: The method quality control and validation that were used have been added in Line (139-148).

Comment: There are some typos.

Response: Authors apologize for the typos. Manuscript has been carefully revised to correct them.

Reviewer 3 Report

Mesa-Infante et al. have made a toxic risk assessment on fluoride exposure of human consuming soybean beverages. This paper is well-thought and well-written. The work carries some important information on the fluoride level in plant-based milk alternative soy milk. However, the following points need to addressed.

1.      Keywords – “toxic risk” should be replaced with “plant-based alternative”

2.      Tables 1-5 – the authors should provide sufficient foot note information to provide the full form of all abbreviations, a brief method used for parameter determination and equations used for obtaining them. For example, ADI, EDI, EU, SD, Max. value, etc.

3.      Section 2.1 – what is the meaning of this statement “This is followed by the material …. fluoride analysis”. Consider rewriting for clarity.

4.      Section 2.3 – include the optimized pH and ionic strength value for maximum sensitivity in determination of fluoride by the method described.

5.      Section 2.3 & 3.1 – It is important to include the calibration curve of fluoride prepared in the concentration range 10-5 to 10-2 M. This figure should include within the linear regression equation and R2 value. This provides the authenticity for the high fluoride levels reported for brands A-C.

6.      Eq.1 & 2 – the dot symbol should be replaced with multiply symbol in both equations.

7.      Section 3 – It is better to provide the nutritional data/profile of brands A, B and C.

Author Response

Comment: Mesa-Infante et al. have made a toxic risk assessment on fluoride exposure of human consuming soybean beverages. This paper is well-thought and well-written. The work carries some important information on the fluoride level in plant-based milk alternative soy milk.

Response: Authors thank the reviewer for their kind words and interest.

Comment: Keywords – “toxic risk” should be replaced with “plant-based alternative”

Response: The keyword has been replaced.

Comment: Tables 1-5 – the authors should provide sufficient foot note information to provide the full form of all abbreviations, a brief method used for parameter determination and equations used for obtaining them. For example, ADI, EDI, EU, SD, Max. value, etc.

Response: Authors thank the reviewer their suggestion, however, as the abbreviations are followed by the full name the first time they appear in the paper, authors believe that adding them to the tables will only made the reading of the manuscript more tedious. The equation for EDI and ADI percentage of contribution are already in the text as eq1 and eq2 (Line XXX), and therefore the same criteria as the abbreviation could be applied.

Comment: Section 2.1 – what is the meaning of this statement “This is followed by the material …. fluoride analysis”. Consider rewriting for clarity.

Response: Authors thanks the reviewer for their suggestion and the statement have been removed from the text.

Comment: Section 2.3 – include the optimized pH and ionic strength value for maximum sensitivity in determination of fluoride by the method described.

Response: The pH range in which the fluoride ISE could work have been added (Line 125). However, its optimal ionic strength is not registered in the electrode technical sheet.

Comment: Section 2.3 & 3.1 – It is important to include the calibration curve of fluoride prepared in the concentration range 10-5 to 10-2 M. This figure should include within the linear regression equation and R2 value. This provides the authenticity for the high fluoride levels reported for brands A-C.

Response: Authors agree with the reviewer and therefore one of the calibration curves obtained have been included in section 2.3.

Comment: Eq.1 & 2 – the dot symbol should be replaced with multiply symbol in both equations.

Response: The symbol has been replaced.

Comment: Section 3 – It is better to provide the nutritional data/profile of brands A, B and C.

Response: Authors thank the reviewer for his suggestion and interest. However, as the aim of this paper is to assess fluoride exposure due to soy beverage consumption and its risk assessment, the authors believe that including the nutritional profile does not serve the purpose of this paper and therefore it is not suitable for inclusion.
